# RaPP: Novelty Detection with Reconstruction along Projection Pathway

**Ki Hyun Kim, Sangwoo Shim, Yongsub Lim, Jongseob Jeon,**
**Jeongwoo Choi, Byungchan Kim, Andre S. Yoon**
MakinaRocks
{khkim, sangwoo, yongsub, jongseob.jeon}@makinarocks.ai
{jeongwoo, kbc8894, andre}@makinarocks.ai

## Abstract

We propose RaPP, a new methodology for novelty detection by utilizing hidden space activation values obtained from a deep autoencoder. Precisely, RaPP compares input and its autoencoder reconstruction not only in the input space but also in the hidden spaces. We show that if we feed a reconstructed input to the same autoencoder again, its activated values in a hidden space are equivalent to the corresponding reconstruction in that hidden space given the original input. We devise two metrics aggregating those hidden activated values to quantify the novelty of the input. Through extensive experiments using diverse datasets, we validate that RaPP improves novelty detection performances of autoencoder-based approaches. Besides, we show that RaPP outperforms recent novelty detection methods evaluated on popular benchmarks.

## 1 Introduction

How can we characterize *novelty* when only *normality* information is given? Novelty detection is the mechanism to decide whether a data sample is an outlier with respect to the training data. This mechanism is especially useful in situations where a proportion of detection targets is inherently small. Examples are fraudulent transaction detection (Pawar et al., 2014; Porwal & Mukund, 2018), intrusion detection (Lee, 2017; Aoudi et al., 2018), video surveillance (Ravanbakhsh et al., 2017; Xu et al., 2015b), medical diagnosis (Schlegl et al., 2017; Baur et al., 2018) and equipment failure detection (Kuzin & Borovicka, 2016; Zhao et al., 2017; Beghi et al., 2014). Recently, deep autoencoders and their variants have shown outstanding performances in finding compact representations from complex data, and the reconstruction error has been chosen as a popular metric for detecting novelty (An & Cho, 2015; Vasilev et al., 2018). However, this approach has a limitation of measuring reconstruction quality only in an input space, which does not fully utilize hierarchical representations in hidden spaces identified by the deep autoencoder.

In this paper, we propose RaPP, a new method of detecting novelty samples exploiting hidden activation values in addition to the input values and their autoencoder reconstruction values. While ordinary reconstruction-based methods carry out novelty detection by comparing differences between input data before the input layer and reconstructed data at the output layer, RaPP extends these comparisons to hidden spaces. We first collect a set of hidden activation values by feeding the original input to the autoencoder. Subsequently, we feed the autoencoder reconstructed input to the autoencoder to calculate another set of activation values in the hidden layers. This procedure does not need additional training of the autoencoder. In turn, we quantify the novelty of the input by aggregating these two sets of hidden activation values. To this end, we devise two metrics. The first metric measures the total amount of reconstruction errors in input and hidden spaces. The second metric normalizes the reconstruction errors before summing up. Note that RaPP falls back to the ordinary reconstruction-based method if we only aggregate input values before the input layer and the reconstructed values at the output layer.

Also, we explain the motivations that facilitated the development of RaPP. We show that activation values in a hidden space obtained by feeding a reconstructed input to the autoencoder are equivalent to the corresponding reconstruction in that hidden space for the original input. We refer the latter

quantity as a *hidden reconstruction* of the input. Note that this is a natural extension of the reconstruction to the hidden space. Unfortunately, we cannot directly compute the hidden reconstruction as in the computation of the ordinary reconstruction because the autoencoder does not impose any correspondence between encoding-decoding pairs of hidden layers during the training. Nevertheless, we show that it can be computed by feeding a reconstructed input to the autoencoder again. Consequently, RAPP incorporates hidden reconstruction errors as well as the ordinary reconstruction error in detecting novelty.

With extensive experiments, we demonstrate using diverse datasets that our method effectively improves autoencoder-based novelty detection methods. In addition, we show by evaluating on popular benchmark datasets that RAPP outperforms competing methods recently developed.

Our contributions are summarized as follows.

- We propose a new novelty detection method by utilizing hidden activation values of an input and its autoencoder reconstruction, and provide aggregation functions for them to quantify novelty of the input.

- We provide motivation that RAPP extends the reconstruction concept in the input space into the hidden spaces. Precisely, we show that hidden activation values of a reconstructed input are equivalent to the corresponding hidden reconstruction of the original input.

- We demonstrate that RAPP improves autoencoder-based novelty detection methods in diverse datasets. Moreover, we validate that RAPP outperforms recent novelty detection methods on popular benchmark datasets.

## 2 RELATED WORK

Various novelty detection methods with deep neural networks rely on the reconstruction error (Sakurada & Yairi, 2014; Hoffmann, 2007; An & Cho, 2015), because discriminative learning schemes are not suitable for highly class-imbalanced data which is common in practice. Unsupervised and semi-supervised learning approaches handle such imbalance by focusing on the characterization of normality and detecting samples out of the normality.

Variational Autoencoders (VAE) (Kingma & Welling, 2014) were reported to outperform vanilla autoencoders for novelty detection based on reconstruction error (An & Cho, 2015). To carry out the novelty detection outlined in this approach, an autoencoder needs to be trained only with normal data. The autoencoder encodes the training data, which comprises of only normal data in this case, into a lower-dimensional space and decodes them to the input space. To test novelty, an input value is fed to the autoencoder to produce a reconstructed value and calculate the distance between the input and reconstructed values. This distance is the reconstruction error. A higher reconstruction error means that the input value cannot be encoded onto the lower-dimensional space that represents normal data. Therefore, the input value can be marked as a novelty if its reconstruction error exceeds a certain threshold.

Instead of autoencoders, Generative Adversarial Networks (GAN) have been also suggested to model a distribution of normal data (Sabokrou et al., 2018; Schlegl et al., 2017). Despite the same purpose of discovering a simpler, lower-dimensional representation, the training criterion for GAN is focusing on the quality of data generation rather than the reconstruction quality of training data. Recently, several pieces of research have combined autoencoders and adversarial learning to meet both criteria in dimension reduction and data generation (Haloui et al., 2018; Pidhorskyi et al., 2018; Zenati et al., 2018). One limitation of these methods based on the ordinary reconstruction error is that they do not exploit all the information available along the projection pathway of deep autoencoders. We will explain how to leverage this information for novelty detection in the next section.

From the viewpoint of the diversity and ratio of the normal data in novelty detection, there are two cases available. The first case is when a small fraction of classes are normal. This case has been studied in a one-class classification context, and usually evaluated by organizing training data into a collection of samples belonging to a small number of normal classes (Ruff et al., 2018; Perera & Patel, 2018; Sabokrou et al., 2018; Golan & El-Yaniv, 2018). The second case is when a majority of classes are assigned as normal (An & Cho, 2015; Schlegl et al., 2017; Haloui et al., 2018; Zenati et al., 2018). In this case, normal data is more diverse, and the training data is consist of samples of a

relatively large number of normal classes: e.g., nine digits of MNIST. One setup does not dominate the other, but depending on applications, either can be more suitable than the other. Different methods may perform differently in both cases. In this paper, we evaluate RAPP and other competing methods with experiments in both setups.

## 3 PROPOSED METHOD: RAPP

In this section, we describe the proposed novelty detection method RAPP based on an autoencoder. The main idea is to compare hidden activations of an input and its hidden reconstructions along the projection pathway of the autoencoder. To be precise, we project the input and its autoencoder reconstruction onto the hidden spaces to obtain pairs of activation values, and aggregate them to quantify the novelty of the input. For the aggregation, we present two metrics to measure the total amount of difference within each pair.

### 3.1 RECONSTRUCTION BASED NOVELTY DETECTION

An autoencoder $A$ is a neural network consisting of an encoder $g$ and a decoder $f$, responsible for dimension reduction and its inverse mapping to the original input space, respectively: i.e. $A = f \circ g$. For the purpose, training the autoencoder aims to minimize difference between its input $x$ and output $A(x)$. The space that the encoder $g$ constitutes is called the latent space, and provides more concise representation for data than the input space.

Due to this unsupervised representation learning property, the autoencoder has been widely used for novelty detection. Specifically, training an autoencoder on normal data samples, novelty of a test sample $x$ is measured by the following reconstruction error $\epsilon$:

$$\epsilon = \|x - A(x)\|_2.$$

The test sample $x$ is more likely to be novel as the error $\epsilon(x)$ becomes larger, because it means that $x$ is farther from the manifold that the autoencoder describes.

Although this approach has shown promising results in novelty detection, the reconstruction error alone does not fully exploit information provided by a trained autoencoder especially when its architecture is deep. In other words, hierarchical information identified by the deep architecture is being ignored. This is rather unfortunate because hierarchical representation learning is one of the most successfully proven capabilities of deep neural networks.

To fully leverage that capability, below we will describe the way to exploit hidden spaces to capture the difference between normal and novel samples in more detail.

### 3.2 RECONSTRUCTION ERROR IN HIDDEN SPACES

Let $A = f \circ g$ be a trained autoencoder where $g$ and $f$ are an encoder and a decoder, and $\ell$ be the number of hidden layers of $g$. Namely, $g = g_\ell \circ \cdots \circ g_1$. We define partial computation of $g$ as follows:

$$g_{:i} = g_i \circ \cdots \circ g_1,$$

for $1 \leq i \leq \ell$.

Let $x$ be an input vector, and $\hat{x}$ be its reconstruction by $A$: i.e., $\hat{x} = A(x)$. In addition to comparing $x$ and $\hat{x}$ in the input space, as the ordinary approach does, we examine them in hidden spaces along a projection pathway of $A$. More precisely, feeding $x$ and $\hat{x}$ into $A$, we obtain pairs $(h_i, \hat{h}_i)$ of their hidden representations where

$$h_i(x) = g_{:i}(x),$$
$$\hat{h}_i(x) = g_{:i}(\hat{x}) = g_{:i}(A(x)).$$

Figure 1a illustrates the procedure of computing $h_i$ and $\hat{h}_i$. As a result, novelty of the sample $x$ is quantified by aggregating $H(x) = \{(h_i(x), \hat{h}_i(x)) : 1 \leq i \leq \ell\}$.

The overall procedure of RAPP is summarized in Algorithm 1. To clearly state the required variables to construct $H$, we write the algorithm with the for loop in Lines 3–5, but in practice, all of them

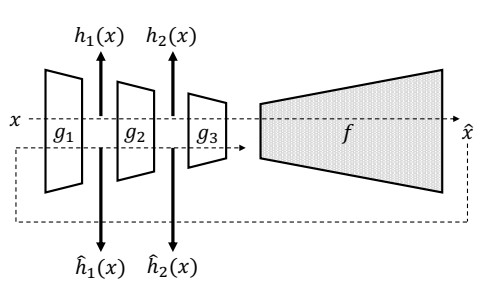 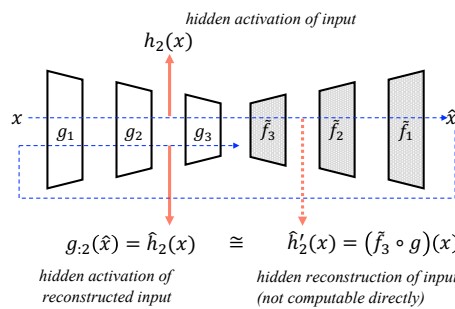

(a) Hidden activations of input and its reconstruction  (b) Indirect computation of hidden reconstruction

Figure 1: (a) Computation of $h_i(x)$ and $\hat{h}_i(x)$. The reconstruction $\hat{x}$ is fed to the same autoencoder that produced itself. (b) Motivation of RAPP. The quantity that RAPP computes, the hidden activation of the reconstruction input, is equivalent to the hidden reconstruction of the input. If $\tilde{f} = f$, computing $\hat{h}'_2(x) = \hat{h}_2(x)$ does not require explicitly evaluating $\tilde{f}_i$ but only $g_i$ and $f = \tilde{f}$.

---

**Algorithm 1:** RAPP to compute a novelty score.

**Input** : Sample $x$, trained autoencoder $A = f \circ g$, the number of layers $\ell$, and aggregation $s$.
**Output:** Novelty score $S$.

1 $\hat{x} = A(x)$.
2 $H = \varnothing$.
3 **foreach** $i$ *in* $1$ *to* $\ell$ **do**
4 $\quad$ $H = H \cup \{(g_{:i}(x), g_{:i}(\hat{x}))\}$.
5 **end**
6 $S = s(H)$.

---

can be computed by feed-forwarding one time each of $x$ and $\hat{x}$ to $g$. Note that RAPP is indeed a generalization of the ordinary reconstruction method with defining $g_0$ as the identity function and $s_{ord}$ as follows.

$$s_{ord}(H(x)) = \|h_0(x) - \hat{h}_0(x)\|_2^2,$$

where $h_0(x) = g_0(x) = x$ and $\hat{h}_0(x) = g_0(\hat{x}) = \hat{x}$.

In this paper, we provide two metrics $s_{SAP}$ and $s_{NAP}$ which more extensively utilize $H$ than $s_{ord}$. Those are especially suited when no prior knowledge exists for the selection of layers to derive a novelty metric, which commonly happens when modeling with deep neural networks. Note that, however, more elaborate metrics can be designed if we have knowledge on or can characterize the spaces.

### 3.2.1 SIMPLE AGGREGATION ALONG PATHWAY (SAP)

This is the most straightforward metric that one can define on $H$. For a data sample $x$, SAP is defined by summing the square of Euclidean distances for all pairs in $H$:

$$s_{SAP}(x) = \sum_{i=0}^{\ell} \|h_i(x) - \hat{h}_i(x)\|_2^2 = \|\mathbf{h}(x) - \hat{\mathbf{h}}(x)\|_2^2,$$

where $\mathbf{h}(x)$ and $\hat{\mathbf{h}}(x)$ are the concatenations of $[h_0(x), \cdots, h_\ell(x)]$ and $[\hat{h}_0(x); \cdots; \hat{h}_\ell(x)]$, respectively.

### 3.2.2 NORMALIZED AGGREGATION ALONG PATHWAY (NAP)

Although SAP is intuitive, it does not consider properties of hidden spaces; distance distributions of pairs in $H$ may be different depending on the individual hidden spaces. For instance, the magnitude

of distances can depend on layers, or there may exist correlated neurons even across layers which are unintentionally emphasized in SAP. To capture clearer patterns, we propose to normalize the distances via two steps: *orthogonalization* and *scaling*.

Let $\mathbf{d}(x) = \mathbf{h}(x) - \hat{\mathbf{h}}(x)$; given a training set $X$, let $\mathbf{D}$ be a matrix whose $i$-th row corresponds to $\mathbf{d}(x_i)$ for $x_i \in X$, and $\bar{\mathbf{D}}$ be the column-wise centered matrix of $\mathbf{D}$. For the normalization, we compute $\bar{\mathbf{D}} = U\Sigma V^\top$, SVD of $\bar{\mathbf{D}}$, to obtain its singular values $\Sigma$ and right singular vectors $V$. For a given data sample $x$, we define $s_{NAP}$ as follows:

$$s_{NAP}(x) = \| \left( \mathbf{d}(x) - \mu_X \right)^\top V\Sigma^{-1} \|_2^2,$$

where $\mu_X$ is the column-wise mean of $D$, and $\mathbf{d}(x)$ is expressed as a column vector. Note that $s_{NAP}$ is equal to the Mahalanobis distance with the covariance matrix $V\Sigma\Sigma V^\top$. Although SVD computation time is quadratic in the number of columns of the target matrix, we observe that its impact is relatively small in practical setups. See Appendix A for more details.

## 4 MOTIVATION OF RAPP

One natural question in using the ordinary reconstruction method is as follows: why do we investigate only the input space? Or, why do we not use information in hidden spaces? While the reconstruction error in the input space is extensively employed, any similar concept does not exist in hidden spaces. One reason is that the corresponding encoding and decoding layers are not guaranteed to express the same space: e.g. permuted dimensions. This is because the autoencoder objective does not have any term involving activations from intermediate hidden layers. As a result, $f_{\ell:i+1}(g(x))$ cannot be considered a reconstruction of $g_{:i}(x)$, except for $i = 0$ with which they become the ordinary reconstruction of and input to an autoencoder, respectively.

Nevertheless, in this section, we will show that there is an indirect way to compute the hidden reconstruction. Precisely, we will show that $\hat{h}_i(x) = g_{:i}(A(x))$ is indeed equivalent to a reconstruction of $g_{:i}(x)$. The overall mechanism is depicted in Figure 1b.

### 4.1 COMPUTATION OF HIDDEN RECONSTRUCTION

Let $A = f \circ g$ be a trained autoencoder, and $M_0 = \{A(x) : x \in \mathbb{R}^n\}$ be the low dimensional manifold that $A$ describes (Pidhorskyi et al., 2018): i.e.,

$$\forall x \in M_0, \ \ x = A(x).$$

Defining $M_i = \{g_{:i}(x) : x \in M_0\}$, which is the low dimensional image of $M_0$ defined by $g_{:i}$, $g$ and $f$ restricted on $M_0$ and $M_\ell$, respectively, are inverse functions of each other.

**Quantifying Hidden Reconstruction**   We first assume that there exists a decoder $\tilde{f} = \tilde{f}_1 \circ \cdots \circ \tilde{f}_\ell$ such that

$$\forall x \in M_\ell, \ \tilde{f}(x) = f(x), \tag{1}$$

$$\forall a \in M_i, \ a = (g_i \circ \tilde{f}_i)(a). \tag{2}$$

The second condition makes $\tilde{f}_{\ell:i+1}$ a proper decoder corresponding to $g_{i+1:}$, and thus, $\tilde{f}$ enables to define the $i$-th hidden reconstruction $\hat{h}'_i(x)$ as follows:

$$\hat{h}'_i(x) = (\tilde{f}_{\ell:i+1} \circ g_{i+1:})(h_i(x)).$$

Finally, we conclude that $\hat{h}'_i(x)$ is equal to $\hat{h}_i(x)$ for $x \in M_0$ as follows.

$$\hat{h}'_i(x) = (\tilde{f}_{\ell:i+1} \circ g_{i+1:})(h_i(x)) = (\tilde{f}_{\ell:i+1} \circ g)(x)$$
$$= (g_{:i} \circ \tilde{f} \circ g)(x) \qquad \text{(by equation 2)}$$
$$= (g_{:i} \circ A)(x) = h_i(\hat{x}) = \hat{h}_i(x). \qquad \text{(by equation 1)}$$

where we do not need $\tilde{f}_i$ for computing $\hat{h}'_i(x)$, but only $g_i$ and $f$. Note that for $x \in M_0$ already on the manifold, its $i$-th hidden reconstruction $\hat{h}'_i(x)$ becomes equal to its corresponding hidden input $h_i(x) = \hat{h}_i(x)$ for every $1 \le i \le \ell$: i.e. $h_i(x) = \hat{h}'_i(x)$ as $x = A(x)$. For $x \notin M_0$, its hidden reconstruction $\hat{h}'_i(x)$ will differ from the input $h_i(x)$.

Table 1: Description of datasets used in our evaluation.

| Name | # Samples | # Features | # Class | Domain | Novelty Target |
|---|---|---|---|---|---|
| MI-F | 25,286 | 58 | 2 | CNC milling | Machine not completed |
| MI-V | 23,125 | 58 | 2 | CNC milling | Workpiece out-of-spec |
| EOPT | 90,515 | 20 | 2 | Storage system | System failures |
| NASA | 4,687 | 33 | 2 | Astronomy | Hazardous asteroids |
| RARM | 20,221 | 6 | 2 | Robotics | Malfunctions |
| STL | 1,941 | 27 | 7 | Steel | Surface defects |
| OTTO | 61,878 | 93 | 9 | E-commerce | Types of products |
| SNSR | 58,509 | 48 | 11 | Electric Currents | Defective conditions |
| MNIST | 70,000 | 784 | 10 | Hand written digits | Digits |
| F-MNIST | 70,000 | 784 | 10 | Fashion articles | Articles |

**Existence of $\tilde{f}$**  Since $x = A(x)$ for $x \in M_0$, $g_i$ and $f_i$ are one-to-one functions from $M_{i-1}$ and $M_i$, respectively. Let us define $\tilde{f}_i = g_i^{-1}$ for $M_i$; then it also holds $\tilde{f} = g^{-1}$ for $M_\ell$. This implies $x = (\tilde{f} \circ g)(x)$ for $x \in M_0$, and consequently, $\tilde{f} = f$ on $M_\ell$. This definition of $\tilde{f}_i$ satisfies the two conditions above, and as discussed, we are able to compute hidden reconstructions given an input $x$, through computing the $i$-th hidden activation of the reconstructed input: i.e. $\hat{h}'_i(x) = (g_{:i} \circ A)(x) = \hat{h}_i(x)$.

**Existence of $\tilde{f}$ with Neural Networks**  Given $g_i$, if the symmetric architecture for $\tilde{f}_i$ is used, we may not be able to learn $\tilde{f}_i = g_i^{-1}$. Neural networks are, however, highly flexible frameworks in which we can deal with models of arbitrary function forms by adjusting network architecture. This property enables us to design a layer capable of representing $\tilde{f}_i$. For instance, even if $\tilde{f}_i$ is too complicated to be represented with a single fully connected layer, we can still approximate $\tilde{f}_i$ by stacking multiple layers. Hence, given $g_i$, $\tilde{f}_i$ can be represented by neural networks.

## 5  EVALUATION

In this section, we evaluate RAPP in comparison to existing methods. To this end, we tested the methods on several benchmarks and diverse datasets collected from Kaggle and the UCI repository which are suitable for evaluating novelty detection methods.

### 5.1  DATASETS AND PROBLEM SETUPS

The datasets from Kaggle and the UCI repository are chosen from problem sets of anomaly detection and multi-class classification, summarized in Table 1. We note that MI-F and MI-V share the same feature matrix, but are considered to be different datasets because their labels *normal* and *abnormal* are assigned by different columns: i.e. *machine completed* and *pass visual inspection*, respectively. We use these datasets to compare RAPP with standard autoencoder-based methods described in Section 5.2.

To compare RAPP with novelty detection methods in recent literatures, we also use popular benchmark datasets for evaluating deep learning techniques: MNIST (LeCun & Cortes, 2010) and F-MNIST (Xiao et al., 2017). For theses datasets, we do not take pre-split training and test sets, but instead merge them for post-processing.

Novelty detection detects novel patterns by focusing on deviations from model-learned normal patterns. Thus, training sets contain only normal samples and test sets contain both normal and anomaly samples in our evaluation setups. Precisely, if a dataset contains an anomaly label, we assign all samples with that label to the test set for detection. If a dataset does not have any anomaly labels, we consider the following two setups.

- Multimodal Normality: A single class is chosen to be the novelty class and the remaining classes are assigned as the normal class. This setup is repeated to produce sub-datasets

with all possible novelty assignments. For instance, MNIST results in a set of datasets with 10 different novelty classes.

- Unimodal Normality: In contrast to the multimodal normality setup, we take one class for normality, and the others for novelty. For instance, MNIST results in a set of datasets with 10 different normal classes.

We applied these two setups to STL, OTTO, SNSR, MNIST, and F-MNIST datasets.

## 5.2 COMPARISON METHOD

We compare RAPP and the other methods using Area Under Receiver Operating Characteristic (AUROC). Note that we do not employ thresholding-based metrics such as $F1$ score because access to abnormal samples is only allowed in testing time. Hence, we focus on the separability of models for novelty with AUROC.

For the datasets in Table 1, we compare the effectiveness of the reconstruction error, SAP and NAP for three models: Autoencoder (AE), Variational Autoencoder (VAE), Adversarial Autoencoder (AAE) (Makhzani et al., 2016). For the benchmark datasets, recent approaches including OCNN (Chalapathy et al., 2018), GPND (Pidhorskyi et al., 2018), DSVDD (Ruff et al., 2018) and GT (Golan & El-Yaniv, 2018) are available. To obtain the performances of the existing approaches, we downloaded their codes and applied against our problem setups.

For MNIST and F-MNIST, we create test sets of novelty ratios 35% for the multimodal setup and 50% for the unimodal setup, where novelty samples in the test sets are randomly selected from given novelty classes. For the other datasets, we take all samples in given novelty classes to create test sets. Note that the expectation value of AUROC is invariant to the novelty ratio.

## 5.3 IMPLEMENTATION DETAILS

We use symmetric architecture with fully-connected layers for the three base models, AE, VAE, and AAE. Each encoder and decoder has 10 layers with different bottleneck size. For the Kaggle and UCI datasets, we carry out PCA for each dataset first. The minimum number of principal components that explain at least 90% of the variance is selected as the bottleneck size of the autoencoders. We set bottleneck size to 20 for benchmark datasets. Leaky-ReLU (Xu et al., 2015a) activation and batch normalization (Ioffe & Szegedy, 2015) layers are appended to all layers except the last layer.

We train AE, VAE and AAE with Adam optimizer (Kingma & Ba, 2015), and select the model with the lowest validation loss as the best model. For training stability of VAE, 10 Monte Carlo samples were averaged in the reparamterization trick (Kingma & Welling, 2014) to obtain reconstruction from the decoder. In the calculation of SAP and NAP, we excluded reconstructions in the input space for MNIST and F-MNIST.

## 5.4 RESULTS

Each AUROC score is obtained by averaging AUROC scores from multiple trials to reduce the random errors in training neural networks: 5 trials for MNIST and F-MNIST, and 20 trials for the other datasets. More results are provided in Appendix: standard deviations in Appendix B, comparison to baselines other than autoencoder variants C, and the effect of varying hidden layers involved in RAPP computation in Appendix D.

### 5.4.1 COMPARISON WITH BASELINES

Table 2 summarizes the result of our performance evaluation; the best score for each model is in bold, and the best score for each dataset with an underline. Since STL, OTTO, SNSR, MNIST, and F-MNIST do not have anomaly labels, their scores are averaged over all possible anomaly class assignments. For instance, the AUROC value for OTTO in the unimodal normality setup is the average of 9 AUROC values with different novelty class assignments.

In Table 2, RAPP shows the highest AUROC scores for most of the cases. If we examine the performance for each dataset, RAPP achieves the best for 10 cases out of 15 (see the underlines).

Table 2: AUROC of RAPP and the baselines.

| Dataset | AE | | | VAE | | | AAE | | |
|---|---|---|---|---|---|---|---|---|---|
| | Recon | SAP | NAP | Recon | SAP | NAP | Recon | SAP | NAP |
| Multimodal Normality | | | | | | | | | |
| STL | **0.723** | 0.703 | 0.711 | 0.700 | 0.675 | **0.711** | 0.726 | 0.711 | 0.724 |
| OTTO | 0.617 | 0.616 | **0.665** | 0.630 | 0.631 | **0.649** | 0.617 | 0.618 | **0.665** |
| SNSR | 0.613 | 0.611 | **0.614** | 0.608 | 0.620 | 0.658 | 0.612 | 0.608 | **0.609** |
| MNIST | 0.825 | 0.881 | **0.899** | 0.900 | 0.965 | **0.965** | 0.847 | 0.911 | **0.929** |
| F-MNIST | 0.712 | 0.725 | **0.734** | 0.725 | 0.728 | **0.755** | 0.721 | 0.710 | **0.727** |
| Unimodal Normality | | | | | | | | | |
| MI-F | 0.607 | 0.670 | **0.705** | 0.591 | 0.572 | **0.678** | 0.632 | 0.692 | **0.704** |
| MI-V | 0.897 | 0.898 | **0.907** | 0.845 | 0.833 | **0.903** | 0.895 | 0.891 | **0.904** |
| EOPT | 0.610 | 0.607 | **0.625** | 0.675 | 0.634 | 0.596 | 0.606 | 0.603 | **0.634** |
| NASA | **0.719** | 0.702 | 0.692 | 0.738 | 0.714 | 0.719 | **0.712** | 0.695 | 0.688 |
| RARM | 0.687 | 0.674 | 0.686 | 0.587 | 0.565 | **0.648** | 0.664 | 0.675 | **0.675** |
| STL | **0.870** | 0.856 | 0.830 | **0.850** | 0.824 | 0.792 | 0.875 | 0.861 | 0.830 |
| OTTO | 0.829 | 0.829 | **0.832** | 0.831 | 0.835 | 0.833 | 0.828 | 0.828 | **0.831** |
| SNSR | 0.979 | 0.982 | **0.990** | 0.975 | **0.979** | 0.964 | 0.979 | 0.983 | **0.991** |
| MNIST | 0.972 | **0.980** | 0.979 | 0.980 | 0.987 | **0.989** | 0.972 | 0.966 | **0.977** |
| F-MNIST | 0.924 | 0.928 | **0.933** | 0.926 | 0.926 | **0.942** | 0.922 | 0.905 | **0.928** |

Table 3: AUROC on benchmark datasets.

| Dataset | OCNN | GPND | DSVDD | GT | $NAP_{AE}$ | $NAP_{VAE}$ | $NAP_{AAE}$ |
|---|---|---|---|---|---|---|---|
| Multimodal Normality (Novelty Ratio: 35%) | | | | | | | |
| MNIST | 0.600 | 0.501 | 0.622 | 0.893 | 0.899 | **0.965** | 0.929 |
| F-MNIST | 0.609 | 0.691 | 0.610 | 0.725 | 0.734 | **0.755** | 0.727 |
| Unimodal Normality (Novelty Ratio: 50%) | | | | | | | |
| MNIST | 0.927 | 0.971 | 0.922 | 0.974 | 0.979 | **0.989** | 0.977 |
| F-MNIST | 0.915 | 0.917 | 0.923 | 0.935 | 0.933 | **0.942** | 0.928 |

### 5.4.2 COMPARISON WITH COMPETITORS

Table 3 summarizes the comparison of RAPP to recent novelty detection methods. As in Table 2, AUROC values are calculated by averaging results from 10 cases with different anomaly class assignments for both datasets.

Except for the unimodal F-MNIST setup, NAP outperforms all competing methods regardless of base model choice. Notably, NAP combined with VAE always shows the best performance, which is even higher than that of GT relying on image-specific data transformations for all cases.

## 6 CONCLUSION

In this paper, we propose a novelty detection method which utilizes hidden reconstructions along a projection pathway of deep autoencoders. To this end, we extend the concept of reconstruction in the input space to hidden spaces found by an autoencoder and present a tractable way to compute the hidden reconstructions, which requires neither modifying nor retraining the autoencoder. Our experimental results show that the proposed method outperforms other competing methods in terms of AUROC for diverse datasets including popular benchmarks.

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

# A    SVD COMPUTATION TIME

We compare running times of training an autoencoder and computing SVD for NAP. We choose two packages for the SVD computation: Pytorch SVD and fbpca provided in `https://fbpca.readthedocs.io/en/latest/.`

Since the time complexity of SVD is linear in the number of data samples[1], we mainly focus on the performance of SVD with varying the number of columns of the input matrix that SVD is applied. To obtain variable sizes of the columns, we vary the depth and bottleneck size of autoencoders.

The result is shown below. Notably, Pytorch SVD utilizing GPU is at least 47x faster than training neural networks. Even, fbpca running only on CPU achieves at least 2.4x speedup.

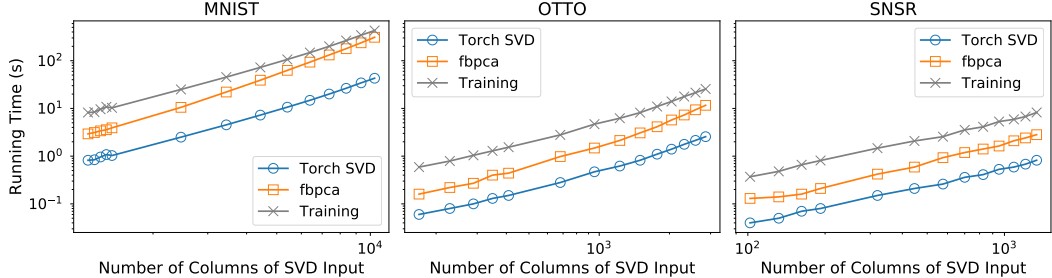

The detailed setups to obtain the matrices for the experiment are given in the table below:

| | MNIST | | OTTO | | SNSR | |
| --- | --- | --- | --- | --- | --- | --- |
| | Depth | Bottleneck Size | Depth | Bottleneck Size | Depth | Bottleneck Size |
| 1 | 20 | 100 | 20 | 40 | 20 | 90 |
| 2 | 18 | 100 | 18 | 40 | 18 | 90 |
| 3 | 16 | 100 | 16 | 40 | 16 | 90 |
| 4 | 14 | 100 | 14 | 40 | 14 | 90 |
| 5 | 12 | 100 | 12 | 40 | 12 | 90 |
| 6 | 10 | 100 | 10 | 40 | 10 | 90 |
| 7 | 8 | 100 | 8 | 40 | 8 | 90 |
| 8 | 6 | 100 | 6 | 40 | 6 | 90 |
| 9 | 4 | 100 | 4 | 40 | 4 | 90 |
| 10 | 2 | 100 | 2 | 40 | 2 | 90 |
| 11 | 2 | 80 | 2 | 30 | 2 | 70 |
| 12 | 2 | 60 | 2 | 20 | 2 | 50 |
| 13 | 2 | 40 | 2 | 10 | 2 | 30 |
| 14 | 2 | 20 | | | 2 | 10 |

---

[1]This is common in practice where the number of data samples is larger than the number of features

# B  STANDARD DEVIATIONS OF EXPERIMENTAL RESULTS

We provide the standard deviations of the result in Table 2. Given a dataset and a setup, multimodal or unimodal, AUROC values are first averaged over multiple cases with different novelty class assignments; then, the standard deviations are calculated for those averaged values over multiple trials. The number of cases and the number of trials depend on datasets, and we refer to Section 5 for more details.

| Dataset | AE Recon | AE SAP | AE NAP | VAE Recon | VAE SAP | VAE NAP | AAE Recon | AAE SAP | AAE NAP |
|---|---|---|---|---|---|---|---|---|---|
| **Multimodal Normality** | | | | | | | | | |
| STL | **0.723** (0.005) | 0.703 (0.007) | 0.711 (0.006) | 0.700 (0.012) | 0.675 (0.018) | **0.711** (0.012) | **0.726** (0.009) | 0.711 (0.011) | 0.724 (0.016) |
| OTTO | 0.617 (0.002) | 0.616 (0.001) | **0.665** (0.004) | 0.630 (0.003) | 0.631 (0.005) | **0.649** (0.003) | 0.617 (0.001) | 0.618 (0.003) | **0.665** (0.004) |
| SNSR | 0.613 (0.002) | 0.611 (0.003) | **0.614** (0.004) | 0.608 (0.001) | 0.620 (0.002) | **0.658** (0.003) | 0.612 (0.005) | 0.608 (0.005) | **0.609** (0.005) |
| MNIST | 0.825 (0.001) | 0.881 (0.007) | **0.899** (0.008) | 0.900 (0.004) | **0.965** (0.005) | **0.965** (0.003) | 0.847 (0.005) | 0.911 (0.006) | **0.929** (0.005) |
| F-MNIST | 0.712 (0.001) | 0.725 (0.004) | **0.734** (0.004) | 0.725 (0.003) | 0.728 (0.002) | **0.755** (0.006) | 0.721 (0.001) | 0.710 (0.005) | **0.727** (0.005) |
| **Unimodal Normality** | | | | | | | | | |
| MI-F | 0.607 (0.023) | 0.670 (0.041) | **0.705** (0.018) | 0.591 (0.013) | 0.572 (0.019) | **0.678** (0.047) | 0.632 (0.028) | 0.692 (0.042) | **0.704** (0.024) |
| MI-V | 0.897 (0.010) | 0.898 (0.005) | **0.907** (0.005) | 0.845 (0.021) | 0.833 (0.032) | **0.903** (0.010) | 0.895 (0.005) | 0.891 (0.006) | **0.904** (0.005) |
| EOPT | 0.610 (0.015) | 0.607 (0.014) | **0.625** (0.008) | **0.675** (0.017) | 0.634 (0.007) | 0.596 (0.002) | 0.606 (0.011) | 0.603 (0.013) | **0.634** (0.010) |
| NASA | **0.719** (0.009) | 0.702 (0.015) | 0.692 (0.016) | **0.738** (0.009) | 0.714 (0.015) | 0.719 (0.009) | **0.712** (0.016) | 0.695 (0.027) | 0.688 (0.024) |
| RARM | **0.687** (0.038) | 0.674 (0.031) | 0.686 (0.030) | 0.587 (0.021) | 0.565 (0.029) | **0.648** (0.022) | 0.664 (0.015) | 0.675 (0.023) | **0.675** (0.040) |
| STL | **0.870** (0.007) | 0.856 (0.007) | 0.830 (0.007) | **0.850** (0.006) | 0.824 (0.003) | 0.792 (0.034) | **0.875** (0.003) | 0.861 (0.004) | 0.830 (0.008) |
| OTTO | 0.829 (0.001) | 0.829 (0.001) | **0.832** (0.001) | 0.831 (0.002) | **0.835** (0.003) | 0.833 (0.004) | 0.828 (0.002) | 0.828 (0.001) | **0.831** (0.002) |
| SNSR | 0.979 (0.001) | 0.982 (0.001) | **0.990** (0.001) | 0.975 (0.003) | **0.979** (0.002) | 0.964 (0.043) | 0.979 (0.001) | 0.983 (0.002) | **0.997** (0.001) |
| MNIST | 0.972 (0.001) | **0.980** (0.001) | 0.979 (0.000) | 0.980 (0.002) | 0.987 (0.002) | **0.989** (0.001) | 0.972 (0.001) | 0.966 (0.004) | **0.977** (0.001) |
| F-MNIST | 0.924 (0.002) | 0.928 (0.002) | **0.933** (0.002) | 0.926 (0.001) | 0.926 (0.001) | **0.942** (0.000) | 0.922 (0.001) | 0.905 (0.002) | **0.928** (0.002) |

# C    COMPARING RAPP TO BASELINES OTHER THAN NEURAL NETWORKS

The table below shows the result of comparing RAPP and baselines not based on neural networks. Note that the baselines are run with hyperparameters or hyperparameter search spaces provided in Ruff et al. (2018), while NAP is not further tuned.

| Dataset | OCSVM | ISOF | PCA | kPCA | $NAP_{AE}$ | $NAP_{VAE}$ | $NAP_{AAE}$ |
|---------|-------|------|-----|------|-----------|------------|------------|
| | | | Multimodal Normality | | | | |
| STL | 0.608 | 0.635 | 0.700 | 0.693 | 0.711 | 0.711 | **0.724** |
| OTTO | 0.628 | 0.548 | 0.572 | 0.644 | **0.665** | 0.649 | **0.665** |
| SNSR | 0.524 | 0.526 | 0.536 | 0.580 | 0.614 | **0.658** | 0.606 |
| MNIST | 0.566 | 0.562 | 0.698 | 0.701 | 0.899 | **0.965** | 0.929 |
| F-MNIST | 0.571 | 0.650 | 0.670 | 0.676 | 0.734 | **0.755** | 0.727 |
| | | | Unimodal Normality | | | | |
| MI-F | 0.777 | 0.826 | 0.546 | **0.838** | 0.705 | 0.678 | 0.704 |
| MI-V | 0.840 | 0.839 | 0.874 | 0.888 | **0.907** | 0.903 | 0.904 |
| EOPT | 0.597 | 0.595 | 0.552 | **0.693** | 0.625 | 0.596 | 0.634 |
| NASA | 0.573 | 0.664 | **0.765** | 0.699 | 0.692 | 0.719 | 0.688 |
| RARM | 0.749 | **0.752** | 0.587 | 0.734 | 0.734 | 0.648 | 0.675 |
| STL | 0.879 | 0.871 | **0.883** | 0.880 | 0.880 | 0.792 | 0.830 |
| OTTO | 0.821 | 0.699 | 0.799 | 0.823 | 0.823 | **0.833** | 0.831 |
| SNSR | 0.959 | 0.893 | 0.894 | 0.970 | 0.970 | 0.964 | **0.991** |
| MNIST | 0.894 | 0.919 | 0.903 | 0.906 | 0.979 | **0.989** | 0.977 |
| F-MNIST | 0.884 | 0.854 | **0.950** | 0.943 | 0.933 | 0.942 | 0.928 |

# D NAP Performance over Increasing Involved Hidden Layers

We investigate the performance of NAP while increasing the number of hidden layers involved in the NAP computation. Specifically, we consider two ways for the increment: 1) adding hidden layers one by one from the input layer (forward addition), and 2) adding hidden layers one by one from the bottleneck layer (backward addition). Experimental results on three datasets are shown below. For most cases, more hidden layers tend to result in higher performance. The values are obtained from one trial without averaging results from multiple trials.

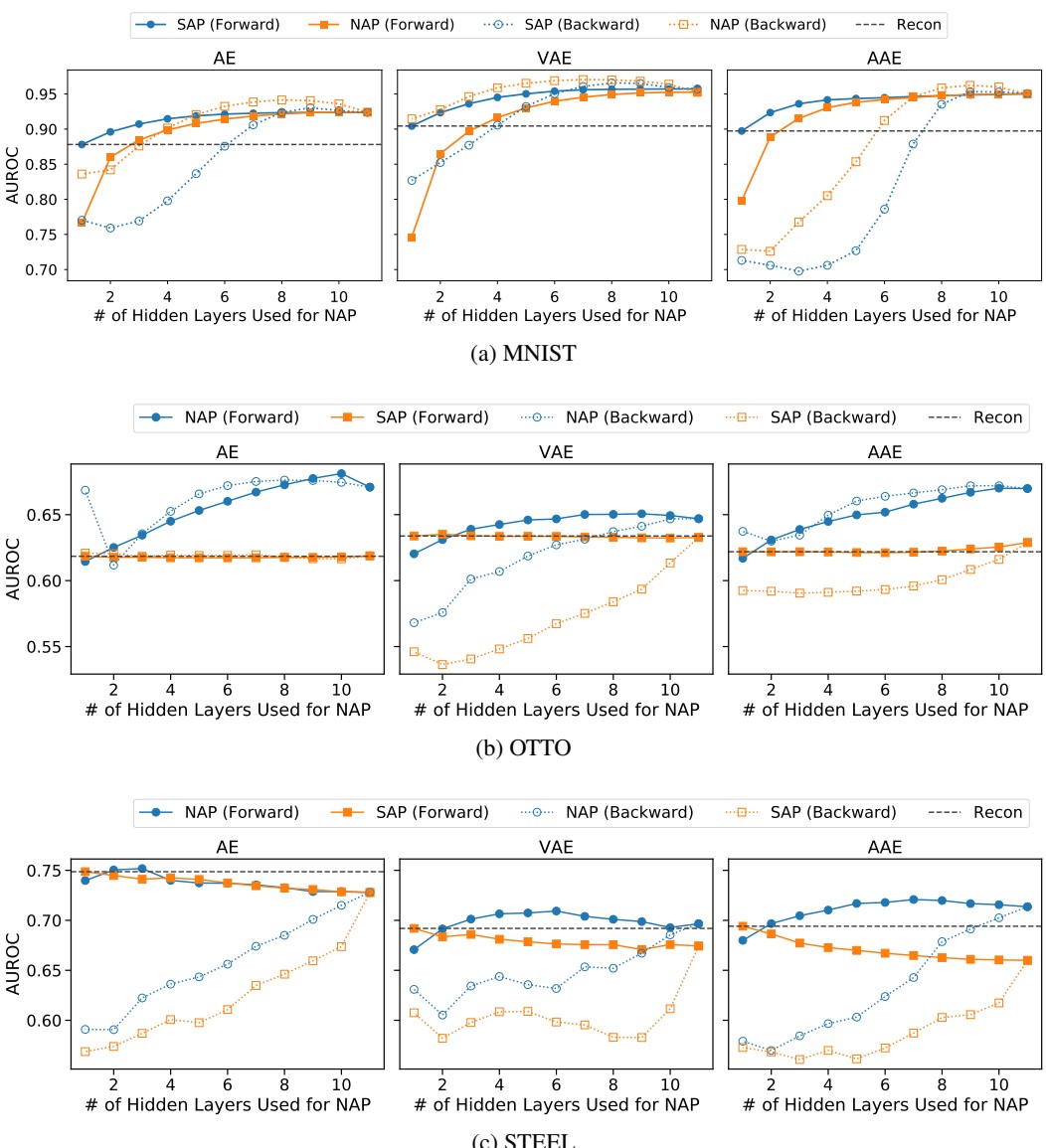

