# OpenReview forum: "RaPP: Novelty Detection with Reconstruction along Projection Pathway"
_ICLR.cc/2020/Conference — Accept (Poster)_

### Official Review · AnonReviewer3 · 2019-10-21
**Official Blind Review #3**

**Rating:** 6

**Review:**

UPDATE:
I acknowledge that I‘ve read the author responses as well as the other reviews.

I appreciate the improvements and clarifications the authors have made, especially adding an ablation study to see the benefits of adding additional layers. I updated my score to Weak Accept (6).

####################

This paper considers deep autoencoders (AEs) for the unsupervised novelty/anomaly detection task and proposes to extend the standard AE anomaly score, given by the reconstruction error between the input and output in the original data space, to also utilize the reconstruction errors of the hidden activations in the AE network. The proposed method, Reconstruction along Projection Pathway (RaPP), specifically compares the hidden activations of all encoder layers given by the original input $x$ with the activations of the same units given by feeding the reconstruction $\hat{x}$ back into the AE. Thus RaPP compares the activation statistics of the original input $x$ and its reconstruction $\hat{x}$ along the encoder projection pathway from original data space to latent code space. Two ways for aggregating those reconstruction errors to a final anomaly score are presented: (1) Simple Aggregation Along Pathway (SAP) which simply computes the sum of reconstruction errors, and (2) Normalized Aggregation Along Pathway (NAP) which computes the sum of reconstruction errors after normalization via Singular Value Decomposition (SVD). The paper conclusively presents experiments on eight datasets from various domains, in which SAP and NAP are compared to the reconstruction error baseline for vanilla AE, VAE, and AAE, as well as experiments on MNIST and Fashion-MNIST in which NAP is compared to state-of-the-art deep anomaly detectors.

Though this work is well presented and indicates promising results, I think the paper should not yet be accepted due to the following main reasons:
(i) The experimental evaluation indicates promising, but not yet convincing results;
(ii) The computational complexity of NAP seems to be a major limitation of RaPP which is not addressed in the text;
(iii) The added value/insights from the theoretical Section 4 (Motivation of RaPP) are not clear.

(i) I think the experimental section shows promising, but not yet convincing results. To judge the significance of results, I think the paper should address the following:
(ia) The experiments on the eight non-image datasets should include other baselines (e.g. OC-SVM, Isolation Forest) besides the standard AE reconstruction error. One should expect SAP and NAP to improve over the standard AE since both methods include the original data space reconstruction errors as well. Moreover, the advantage of deep approaches on such non-image datasets is less clear [7] why a comparison to well-known baselines should be given.
(ib) The main motivation for deep approaches to anomaly detection are large and complex datasets [6, 5, 4, 2]. I think the comparison to recent, state-of-the-art deep competitors should at least include another dataset more complex than MNIST or Fashion-MNIST, e.g. CIFAR-10 as reported in the previous works or MVTec [1].
(ic) I think the proposed method begs for an ablation study of subsequently adding the reconstruction errors of additional layers. This would clearly demonstrate the potential benefits of adding the hidden reconstructions.

(ii) The experiments indicate that a proper normalization of the hidden activation reconstruction errors is crucial for improving detection performance. NAP shows consistent improvements, whereas SAP often performs similar to the AE baseline. However, the current SVD normalization procedure on a matrix with dimensions number of samples × number of hidden encoder units seems extremely costly to me and appears to be a major limitation towards larger datasets or networks. Could you comment on this since this is not yet addressed in the manuscript. Have you tried using Batch Normalization (after activation) together with per-layer averaging? To me, this seems the natural first choice to normalize unit scores and to account for different layer widths. Do you apply SVD on mini-batches?

(iii) The additional insights from the theoretical Section 4 are not clear to me. I think the presented reconstruction property for the hidden layers follows somewhat directly per definition for symmetrically constructed deep autoencoders (specifically if weights would be shared in addition). For a theoretical contribution, on the other hand, the proof and proposition should be fully rigorous in my mind, i.e. stating all the necessary assumptions on the function class (e.g. you implicitly assume invertibility and thus some smoothness of the $g_i$'s which Conv+ReLU modules do not satisfy for instance). As of now, I think this section does not add to intuition, but on the other hand is not completely rigorous. Maybe I am missing something?

The overall presentation of the paper is good (clear writing and structure, polished Figures and Tables). The work is well motivated and properly placed in the literature. Maybe since the approach is rather simple (which I don’t find negative), the author felt the need to add some rigor to the paper, which I think would not be necessary for a significant contribution if the experimental results hold up against the additional baselines and more complex datasets as described in (i).


####################
*Additional Feedback*

*Positive Highlights*
1. Simple idea that does not require autoencoder modification or retraining that indicates improved anomaly detection results.
2. The work is well placed in the literature. The related work includes all relevant and recent major works on the subject matter.
3. I appreciate the evaluation on both anomaly/novelty detection setups, unimodal and multimodal.
4. Comparison to recent OC-NN [3], GPND [5], Deep SVDD [6], and GT [4].
5. The writing, structure and overall presentation is good.

*Ideas for Improvement*
6. Include additional baselines and more complex datasets as described in (i).
7. Address the computational complexity of RaPP as in described in (ii).
8. Maybe cut the methodical/theoretical parts in Section 3.2 and Section 4 a bit. I think they are rather straightforward. Maybe combine Figures 1+2 as well. Extend the experimental evaluation instead.
9. Report the AUROC standard deviations over the trials as well to better infer statistical significance of the results (defer to appendix if space is a constraint).

*Minor comments*
10. Section 2: “Unsupervised and semi-supervised learnings” » “Unsupervised and semi-supervised learning approaches”.
11. Section 2: “Variational Autoencoders (VAE) was reported ...” » “Variational Autoencoders (VAE) were reported ...”
12. Section 3.1: “Due to this representation learning property, the autoencoder has been widely used for novelty detection.” » emphasis on unsupervised learning property, specifically.
13. Section 3.1: “Although this approach has shown a promising result in novelty detection ...” » “Although this approach has shown promising results in novelty detection ...”
14. Section 3.1, last sentence: “... in more details.” » “... in more detail.”
15. Section 3.2: “Those are especially suited for the case of zero-knowledge to interpret identified hidden spaces, which commonly happens when modeling with deep neural networks.” Zero-knowledge case? Reference?
16. In Section 5.1: “Further setups are described in Section 5.1”?
17. Section 5.4.1: “Also, we showed the best score ...” » “Also, we show the best score ...”.
18. Section 5.2: “... maintaining novelty ratios to 35% for the multimodal and 50% for the unimodal normality setups, respectively.” Why use different ratios?


####################
*References*
[1] P. Bergmann, M. Fauser, D. Sattlegger, and C. Steger. Mvtec ad–a comprehensive real-world dataset for unsupervised anomaly detection. In Proceedings of the IEEE Conference on Computer Vision and Pattern Recognition, pages 9592–9600, 2019.
[2] R. Chalapathy and S. Chawla. Deep learning for anomaly detection: A survey. arXiv preprint arXiv:1901.03407, 2019.
[3] R. Chalapathy, A. K. Menon, and S. Chawla. Anomaly detection using one-class neural networks. arXiv preprint arXiv:1802.06360, 2018.
[4] I. Golan and R. El-Yaniv. Deep anomaly detection using geometric transformations. In NIPS, 2018.
[5] S. Pidhorskyi, R. Almohsen, and G. Doretto. Generative probabilistic novelty detection with adversarial autoencoders. In NeurIPS, pages 6822–6833, 2018.
[6] L. Ruff, R. A. Vandermeulen, N. Görnitz, L. Deecke, S. A. Siddiqui, A. Binder, E. Müller, and M. Kloft. Deep one-class classification. In International Conference on Machine Learning, pages 4393–4402, 2018.
[7] L. Ruff, R. A. Vandermeulen, N. Görnitz, A. Binder, E. Müller, K.-R. Müller, and M. Kloft. Deep semi-supervised anomaly detection. arXiv preprint arXiv:1906.02694, 2019.

**Experience Assessment:**

I have published in this field for several years.

**Review Assessment: Checking Correctness Of Derivations And Theory:**

I carefully checked the derivations and theory.

**Review Assessment: Checking Correctness Of Experiments:**

I carefully checked the experiments.

**Review Assessment: Thoroughness In Paper Reading:**

I read the paper thoroughly.

---

> ### Author Response · Authors · 2019-11-10
> **Response to the review**
>
> Thank you for the detailed feedback and ideas for improvement.
> At this time, we are sharing what we've done so far on your ideas.
>
> [(ia) Comparison to well-known baselines]
>
> Thank you for the suggestion. We are evaluating the baselines you suggested, and will add the result in our revision.
> ​
>
> [(ib) Comparison on more complex datasets]​
>
> We are also trying to do additional experiments on a complex image dataset. As soon as we get the result, we will report it as a reply.
> ​
>
> [(ic) Experimental results with subsequently adding hidden layers]
>
> We indeed have experimental results about the comment. We will add the result to our revision.
>
>
> [Cost of SVD]
>
> As you pointed out, SVD takes quite a bit of computation resources: e.g. $nm^2$ for a full SVD in our case. However, since SVD computation is required only at a training phase in RaPP, we are more flexible in utilizing computational resources. To be more efficient, we can also employ probabilistic approximation algorithms [1].
> We will add an explanation about this in our revised paper. Also, we are carrying out experiments to check computational advantage of [1] with the implementation provided in [2], and we will share the results in the next reply.
>
> [1] Halko, N. and Martinsson, P. G. and Tropp, J. A. “Finding Structure with Randomness: Probabilistic Algorithms for Constructing Approximate Matrix Decompositions.” SIAM Rev., 53(2), 217–288, 2011
> [2] https://fbpca.readthedocs.io/en/latest/
>
>
> [(iii) Revision of Section 4]
> ​
> We would like to clarify the motivation in Section 4.
>
> ** Background **
> Let us consider a symmetric autoencoder $A$. In general, its pair of the corresponding encoding and decoding layers is not guaranteed to express the same space: an obvious example is permuted dimensions. This is because training $A$ does not care about activations from intermediate hidden layers. As a result, directly comparing $g_{:i}(x)$ and $f_{\ell:i+1}(g(x))$ does not make sense, except for $i=0$ with which the comparison becomes the same as computing ordinary reconstruction error. This makes the concept of “hidden reconstruction” not defined as done with $A$ for the ordinary reconstruction, though it sounds reasonable.
>
> ** What we showed  **
> Nevertheless, we show that activation vectors in an encoding hidden layer obtained by feeding the original input $x$ and its reconstruction $\hat{x}=A(x)$ to the same network $A$ have the relation of input and reconstruction for the corresponding hidden space. That is, $g_{:i}(A(x))$ is equivalent to a reconstruction for $g_{:i}(x)$ in the $i$-th hidden space of $A$.
>
> ** Assumption **
> For the conclusion above, we only assumed that given a trained autoencoder $A$, $x = A(x)$ for $x\in M_0$ where $M_0$ is a low dimensional manifold in the paper. With this assumption, $g$ restricted on $M_0$ and $f$ restricted on $M_\ell$ must become one-to-one functions. Here, we note that we did not make the statement on training data but on the manifold $M_0$ that the trained autoencoder describes.
>
> In Section 4, we tried to provide what quantity $\hat{h}_i(x)=g_{:i}(A(x))$ means or why it is meaningful in connection to the well-known reconstruction concept.
> Reviewing Section 4 by ourselves, we think readers can be confused about the point. We will make it clearer in our revision.
> ​
>
> [Reporting standard deviations]
> ​
> We will add the result in the appendix of our revised paper.
> ​
>
> [About other feedbacks]
> ​
> We are now working on a revision and will include your feedback.

---

> ### Author Response · Authors · 2019-11-15
> **Additional comments**
>
>
> [(ic) Experimental results with subsequently adding hidden layers]
> ​
> We obtained the results for MNIST and STL datasets, and will add in the appendix. In general, performance tends to get higher as more layers are used.
>
>
> [Cost of SVD]
>
> We conducted experiments to evaluate the scalability of SVD with Pytorch implementation and Facebook’s implementation of a fast randomized algorithm (fcpca) [2]. Note that PyTorch SVD [3] partially utilizes GPU, but fcpca only runs on CPU.
>
> ** Setup **
> The experiments were carried out for MNIST, SNSR, and OTTO which have a large number of samples and high dimensionality as can be seen in Table 1. Since the time complexity of SVD is linear in the number of data samples (because n_samples > data_dim in most of the cases), we mainly tested the performance of SVD across the various depth and bottleneck sizes of networks, which is directly related to the number of columns of the matrix fed to SVD.
>
> ** Result **
> We observed that Pytorch SVD (used in our paper) is faster than fbpca, and takes much shorter than autoencoder training time. In our result, Pytorch SVD and fbpca are at least 47x and 6.5x faster than training an autoencoder, respectively.
> We will add the result in the following revision.
>
> ** Conclusion **
> The impact of the SVD computation (required only at a training phase) in NAP is relatively small compared to training an autoencoder in practical setups.
>
> [1] Halko, N. and Martinsson, P. G. and Tropp, J. A. “Finding Structure with Randomness: Probabilistic Algorithms for Constructing Approximate Matrix Decompositions.” SIAM Rev., 53(2), 217–288, 2011
> [2] https://fbpca.readthedocs.io/en/latest/
> [3] https://pytorch.org/docs/stable/torch.html?highlight=svd#torch.svd
>
>
> [About other feedbacks]
>
> ** Minor Comment 15 **
> We used “zero-knowledge” as a non-technical term to indicate when no prior knowledge exists for the selection of layers to derive a novelty metric. Our approach treats all the layers equally to calculate the metrics. We will revise the statement to avoid confusion in the next revision.
>
>
> ** Minor Comment 18 **
>
> We used as many data samples as possible for every setup, except for MNIST and FMNIST with the unimodality setup that we followed the GPND setup [4]. Since AUROC is invariant in expectation regardless of the proportion of novelty samples, we took the maximum proportion of 50% used in [4].
>
> [4] Stanislav Pidhorskyi, Ranya Almohsen, and Gianfranco Doretto. Generative probabilistic novelty detection with adversarial autoencoders. In NeurIPS, pp. 6823–6834, 2018.
>
>
> ** Other Minor Comments**
> We will revise our paper as suggested by the reviewer.

---

> ### Author Response · Authors · 2019-11-15
> **Additional Comments**
>
>
> [(ia) Comparison to well-known baselines]
>
> The table in Appendix C shows the result of comparing RaPP and baselines not based on deep approaches. With the baseline models, for some datasets best AUROC performance is altered, i.e., baseline models show higher AUROC performance. We’d like to point out, however, that there is room for improvement for the NAP result since hyperparameter tuning (e.g., depth of deep architecture, size of bottleneck, training epochs and etc) was not performed while the baseline models used tuned hyperparameters as provided in the DSVDD paper [1]. The evaluation of the baselines for SNSR, MNIST, and F-MNIST are still on-going. We will include the results in the final revision.
>
> [1] Lukas Ruff, Robert Vandermeulen, Nico Goernitz, Lucas Deecke, Shoaib Ahmed Siddiqui, Alexander Binder, Emmanuel Mu ̈ller, and Marius Kloft. Deep one-class classification. In ICML, 2018.
>
>
> [(ib) Experiments on complex dataset: MVTec AD]
>
> We evaluated RaPP (NAP) on MVTec AD dataset as suggested. For the preprocessing of the dataset, the following procedures were applied. First, each image is grayscaled and the resolution was lowered. Second, the image was segmented to $32\times32$ patches.
> We used VAE with 8 layers for each fully-connected encoder and fully-connected decoder, and trained 200 epochs.  Below is the result of our evaluations.
>
> **Result**
>
> +------------+-----------------+----------------+-----------+
> |  Category  |   Recon AUROC   |   RaPP AUROC   | AE(L2)[1] |
> +------------+-----------------+----------------+-----------+
> | Carpet     | 0.5846+-0.0022  | 0.5612+-0.0105 |      0.59 |
> | Grid       | 0.5444+-0.0039  | 0.7050+-0.0121 |      0.90 |
> | Leather    | 0.5603+-0.0099  | 0.8269+-0.0227 |      0.75 |
> | Tile       | 0.6055+-0.0007  | 0.5387+-0.0032 |      0.51 |
> | Wood       | 0.6794+-0.0008  | 0.7030+-0.0090 |      0.73 |
> | Bottle     | 0.6744+-0.0111  | 0.7602+-0.0025 |      0.86 |
> | Cable      | 0.6711+-0.0142  | 0.6939+-0.0048 |      0.86 |
> | Capsule    | 0.6781+-0.0092  | 0.8192+-0.0231 |      0.88 |
> | Hazelnut   | 0.7524+-0.0011  | 0.7491+-0.0075 |      0.95 |
> | Metal Nut  | 0.4692+-0.0030  | 0.5889+-0.0026 |      0.86 |
> | Pill       | 0.6275+-0.0065  | 0.6860+-0.0102 |      0.85 |
> | Screw      | 0.8140+-0.0036  | 0.7592+-0.0002 |      0.96 |
> | Toothbrush | 0.7286+-0.0099  | 0.8559+-0.0300 |      0.93 |
> | Transistor | 0.5983+-0.0014  | 0.6668+-0.0121 |      0.86 |
> | Zipper     | 0.5318+-0.0334  | 0.6356+-0.0011 |      0.77 |
> +------------+-----------------+----------------+-----------+
> | Average    | 0.6346+-0.0894  | 0.7033+-0.0930 |    0.8173 |
> +------------+-----------------+----------------+-----------+
>
> The results show that AUROC obtained from RaPP in general is higher than AUROC obtained from reconstruction only. Yet it is noted that the overall performance is still lower than the quoted performance in the cited paper [1] except Leather and Tile. In order to make an apple-to-apple comparison, more work is needed to include 1) Incorporating with CNN architecture and 2) Data Augmentation, which is missing in our evaluation. In our evaluation, the resolution was intentionally lowered due to the shortage of time for the evaluation, but this should also be recovered as well to make a fair comparison. We are currently looking into the possibility of extending RaPP to CNN-based models, which will allow rigorous comparisons of RaPP approach with the other existing approaches on more complex image datasets such as CIFAR-10, MVTec and so on.
>
> [1] P. Bergmann, M. Fauser, D. Sattlegger, and C. Steger. Mvtec ad–a comprehensive real-world dataset for unsupervised anomaly detection. In Proceedings of the IEEE Conference on Computer Vision and Pattern Recognition, pages 9592–9600, 2019.

---

> ### Author Response · Authors · 2019-11-15
> **Moved**
>
> The content is moved to the third reply

---

### Official Review · AnonReviewer1 · 2019-10-24
**Official Blind Review #1**

**Rating:** 6

**Review:**

I have read the reviews and the comments.

I appreciate the effort of the authors. I feel positive about the paper and I think it should be accepted.

I confirm my rating.

=================
The paper proposes a new method for novelty detection that is based on measuring the reconstruction error in latent space between layer of the encoder.
The reconstructed sample is fed back to the encoder and activations of the hidden layers of the encoder are compared with the activations that occurred when the original sample was fed into it.

To aggregate the reconstruction error from all layers of the encoder, two methods are proposed SAP (simple aggregation along pathway) and NAP (normalized aggregation along pathway). SAP is simply the sum of the squared norm, while NAP performs decorrelation and normalization of the magnitude.

The idea is novel, well motivated and explained.

It is said in the paper that NAP performs distance normalization by doing orthogonalization and scaling. The way it is described seems to be equivalent to PCA whitening. Thus, the computed distance should be a Mahalanobis distance.

It is not clear why for the VAE case 10 samples are averaged, instead of just using the mean component given by the encoder and passing it to decoder. It is typical to use reparametrization only during training.

Comparison with other state of the art methods is somewhat weak, since only two similar datasets are used (MNIST and F-MNIST).

**Experience Assessment:**

I have published one or two papers in this area.

**Review Assessment: Checking Correctness Of Derivations And Theory:**

I carefully checked the derivations and theory.

**Review Assessment: Checking Correctness Of Experiments:**

I carefully checked the experiments.

**Review Assessment: Thoroughness In Paper Reading:**

I read the paper at least twice and used my best judgement in assessing the paper.

---

> ### Author Response · Authors · 2019-11-10
> **Response to the review**
>
> Thank you for the feedback and the suggestion.
> ​
> [About Mahalanobis distance]
> ​
> They are indeed equivalent. Thank you for pointing this out. If we let the covariance matrix be $S$, in our terminology, $S = \overline{D}^T\overline{D} = V\Sigma \Sigma V^T$.
> Therefore, the mahalanobis distance $(d - \mu)^T S^{-1} (d - \mu) = (d - \mu)^T V\Sigma^{-1} \Sigma^{-1}V^T (d - \mu) = ||\overline{d}^T V\Sigma^{-1}||_2^{2}$, which is the same as $S_{NAP}$.
>
> There was a typo in the manuscript ($d$ must be $\overline{d}$), and we will fix it in our revision. Since our code was correctly written, the numbers in the current manuscript were obtained with the corrected expression above.
>
>
> [About VAE inference]
> ​
> As you pointed out, we carried out experiments with the mean component (‘mu’) given by the encoder during the inference phase and found the results are still consistent within the quoted standard deviation (and potentially better.) We also found that we included the results with K = 1 (K: number of latent samples for reparameterization trick) in the paper instead of K = 10. Thus, we additionally carried out experiments with K = 10, as originally intended, and found the results are also within the quoted standard deviation. See the table below. Evaluations were repeated 5 times to estimate the mean and standard deviation. We will update the paper as well as the code with the results obtained from VAE using mean (‘mu’) for the inference.
>
>
> +------------+-------------+---------------+---------------------+----------------------+---------------------+
> | Dataset | Training | Inference | recon                 | SAP                     | NAP                   |
> +------------+-------------+---------------+----------------------+----------------------+--------------------+
> | MNIST   | k=1          | k=1            | 0.8636+-0.1789 | 0.9070+-0.0779 | 0.9270+-0.0666 |
> |                +-------------+---------------+----------------------+----------------------+--------------------+
> |                | k=10        | k=10         | 0.8965+-0.1687 | 0.9603+-0.0411 | 0.9613+-0.0388 |
> |                +-------------+--------------+-----------------------+----------------------+--------------------+
> |                | k=10        | mu           | 0.9038+-0.1619 | 0.9621+-0.0433 | 0.9654+-0.0348 |
> +------------+-------------+--------------+-----------------------+----------------------+--------------------+
> | FMNIST  | k=1         | k=1          | 0.7101+-0.1379 | 0.6714+-0.1068 | 0.7365+-0.1275 |
> |                +-------------+--------------+-----------------------+----------------------+--------------------+
> |                | k=10        | k=10        | 0.7203+-0.1440 | 0.7320+-0.1258 | 0.7614+-0.1208 |
> |                +-------------+--------------+-----------------------+----------------------+--------------------+
> |                | k=10        | mu           | 0.7244+-0.1439 | 0.7483+-0.1229 | 0.7678+-0.1199 |
> +------------+-------------+---------------+---------------------+----------------------+----------------------+
>
>
> [ About datasets for comparison to other state-of-the-art methods ]
> ​
> We are trying to do additional experiments on a complex image dataset. As soon as we get the result, we will report it as a reply.

---

> ### Author Response · Authors · 2019-11-15
> **Additional Comments**
>
>
> [ About datasets for comparison to other state-of-the-art methods ]
>
> We did experiments to compare RaPP to the state-of-the-art methods for more datasets.
> Please refer to the additional evaluations in the following responses:
> * [Extending Table3 to include other datasets] in the last reply for the first reviewer, and
> * [(ib) Experiments on complex dataset: MVTec AD] in the last reply for the third reviewers

---

### Official Review · AnonReviewer4 · 2019-11-04
**Official Blind Review #4**

**Rating:** 6

**Review:**

This paper proposes a novelty detection method by utilizing latent variables in auto-encoder. Based on this, this paper proposes two metrics to quantifying the novelty of the input. Their main contribution is the NAP metric based on SVD. Their method is empirically demonstrated on several benchmark datasets, and they compare their proposed metrics with other competing methods using AUROC and experiments results are encouraging.

The metrics proposed in this paper are intuitive and interesting. The experiments shown in Table2 is very convincing, and it could be better to extend Table3 to include other datasets (STL,OTTO, etc. )




**Experience Assessment:**

I do not know much about this area.

**Review Assessment: Checking Correctness Of Derivations And Theory:**

I assessed the sensibility of the derivations and theory.

**Review Assessment: Checking Correctness Of Experiments:**

I carefully checked the experiments.

**Review Assessment: Thoroughness In Paper Reading:**

I read the paper at least twice and used my best judgement in assessing the paper.

---

> ### Author Response · Authors · 2019-11-10
> **Response to the review**
>
> Thank you for the feedback.
>
> [Extending Table3 to include other datasets]
> We will answer to your suggestion as soon as ready. We are now examining the applicability of your suggestion.

---

> ### Author Response · Authors · 2019-11-15
> **Additional Comments**
>
>
> [Extending Table3 to include other datasets]
>
> Once again, thank you for your suggestion. In order to make evaluations of the existing methods on the datasets used in Table 1, the following architectural modifications were made for the models in the recent approaches: (1) CNN components were removed and replaced by fully connected (FC) layers. (2) Number of layers and bottleneck size were modified to match that of AE, VAE, AAE in Table 2
>
> The modifications were necessary for the following reasons. First, numeric data has no explicit relation between its features while CNN explicitly utilizes the grid structure of pixels in an image. As such, CNN architecture does not naturally extend to 1D numeric data (not 2D image data). Second, to make the comparison as close as possible, we keep the same number of layers and bottleneck size. Below is the result of the evaluations. We used the same settings for the training as in the cases shown in Table 2. Due to the shortage of time, the evaluation was done only for OCNN and DSVDD in uni- and multi- modal normality cases. GT was excluded since it relies on image-specific data transformations like rotation and flipping.
>
> **Result**
> +---------+------------+-----------+----------+-----------+----------+
> | Dataset |    Type    | OCNN MEAN | OCNN STD | DSVDD MEAN | DSVDD STD |
> +---------+------------+-----------+----------+-----------+----------+
> | STL     | Multimodal |     0.755 |    0.146 |     0.518 |    0.162 |
> | OTTO    | Multimodal |     0.496 |    0.149 |     0.552 |    0.101 |
> | SNSR    | Multimodal |     0.487 |    0.029 |      0.49 |    0.048 |
> |         |            |           |          |           |          |
> | MI-F    | Unimodal   |     0.358 |    0.107 |     0.523 |    0.103 |
> | MI-V    | Unimodal   |      0.43 |    0.061 |      0.46 |    0.042 |
> | EOPT    | Unimodal   |      0.53 |    0.014 |         - |        - |
> | NASA    | Unimodal   |     0.526 |    0.043 |     0.549 |    0.062 |
> | RARM    | Unimodal   |      0.43 |    0.116 |     0.604 |    0.048 |
> | STL     | Unimodal   |     0.467 |    0.094 |     0.566 |    0.124 |
> | OTTO    | Unimodal   |     0.572 |    0.139 |     0.663 |     0.03 |
> | SNSR    | Unimodal   |     0.578 |    0.069 |     0.586 |    0.058 |
> +---------+------------+-----------+----------+-----------+----------+
> (The experiment for DSVDD on EOPT is still on-going)
>
> The result shows AUROC is in general lower than that of AE, VAE and AAE with the reconstruction error. The only exception is OCNN on STL in multimodal normality case where AUROC reaches 0.755. In all cases, AUROC is lower than the highest AUROC among AE, VAE, and AAE results.
>
> We’d like to point out that more study is needed to make the comparison rigorous. In this regard, it would be interesting to see how CNN-based models could extend to incorporate 1D numeric data (e.g., sensors or time-series data). We are currently looking into the possibility of extending RaPP to CNN-based models, which will allow the evaluations of RaPP on more complex image datasets such as CIFAR-10, MVTec and so on.

---

### Author Response · Authors · 2020-02-15
**Revisions in the camera-ready version**

Below, we summarize three main revisions made in the camera-ready version.

1. Removal of the potential statistical bias in the experiments.

We calculated the means and standard deviations for standard scaling from the whole dataset during the preprocessing.
Thus, we carried out the experiments again with the statistics only from the training set to measure the implications of the forward-looking bias.
The updated experimental results indicate that the forward-looking bias does not affect the conclusion of the manuscript.
Hence, we updated only experimental results and retained the conclusion in the camera-ready version.
More specifically, we updated the following parts of the manuscript.

- Table 2 and 3
- Section 5.4.1
- Appendix B, C and D

To obtain statistically more stable results, we increased the number of trials for the tabular datasets.
We updated the corresponding explanations in Section 5.4.

2. Notational change

We updated parts of notations for greater clarity.

- In Figure 1 and its caption, we changed $h_i$ and $\hat{h}$ to functional forms $h_i(x)$ $\hat{h}(x)$, respectively, to increase consistency with the main text.
- In Section 4.1, we changed $h$ to $a$ in Equation (2) to indicate that the entity is an activation output.

3. Fixing explanation of novelty ratios in test sets

We updated details of novelty ratios in our experimental setup for tabular datasets.
See the last paragraph of Section 5.2

---

### Decision · Program_Chairs · 2019-12-19

**Decision:**

Accept (Poster)

**Comment:**

The paper proposes to extend the autoencoder loss in a deep generative model to include per-latent-layer loss terms.  Two variants are proposed: SAP (simple aggregation along pathway) and NAP (normalized aggregation along pathway). SAP is simply the sum of the squared norm, while NAP performs decorrelation and normalization of the magnitude.  This was viewed as novel by the reviewers, and the experiments supported the proposed approach.

In the post rebuttal phase, the inclusion of an ablation study has led to an upgrade in the reviewer recommendation.  As a result, there was a unanimous opinion that the paper is suitable for publication at ICLR.